# Zika virus T-cell based 704/DNA vaccine promotes protection from Zika virus infection in the absence of neutralizing antibodies

**Claude Roth** [1] *, **Bruno Pitard**[2☯], **Laurine Levillayer**[1☯], **Sokchea Lay**[3], **Hoa Thi My Vo**[3,4], **Tineke Cantaert**[3], **Anavaj Sakuntabhai**[1]

**1** Ecology and Emergence of Arthropod-Borne Pathogens Unit, Institut Pasteur, CNRS UMR2000, 75015 Paris, France, **2** Nantes Université, Univ Angers, INSERM, CNRS, Immunology and New Concepts in Immunotherapy, INCIT UMR1232/EMR6001, F-44000 Nantes, France, **3** Immunology Unit, Institut Pasteur du Cambodge, Pasteur Network, Phnom Penh, Cambodia, **4** Oxford University Clinical Research Unit, Ho Chi Minh, Vietnam

☯ These authors contributed equally to this work.
* clroth75@gmail.com

**Data Availability Statement:** All data underlying the findings described in the manuscript are freely available to other researchers, either in a public

## Abstract

Zika virus (ZIKV) and dengue virus (DENV) are closely related flaviviruses co-circulating in the same endemic areas. Infection can raise cross-reactive antibodies that can be either protective or increase risk of severe disease, depending on the infection sequence, DENV serotype and elapsed time between infection. On the contrast, T cell-mediated immunity against DENV and ZIKV is considered protective. Therefore, we have developed a T cell vaccine enriched in immunodominant T cell epitopes derived from ZIKV and evaluated its immunogenicity and efficacy against ZIKV and DENV infection. Mice were vaccinated using DNA vaccine platform using the tetrafunctional amphiphilic block copolymer 704. We show that vaccination of 2 different HLA class I transgenic mice with the ZIKV non-structural (NS) poly-epitope elicits T cell response against numerous ZIKV epitopes. Moreover, vaccination induces a significant protection against ZIKV infection, in the absence of neutralizing or enhancing antibodies against ZIKV. However, vaccination does not induce a significant protection against DENV2. In contrast, immunization with a DENV1-NS poly-epitope induces a significant protection against both DENV1 and DENV2, in the absence of humoral immunity. Taken together, we have shown that T-cell based vaccination could protect against multiple flavivirus infections and could overcome the complexity of antibody-mediated enhancement.

## Author summary

Dengue virus (DENV) and Zika virus (ZIKV) are two closely related flaviviruses transmitted by *Aedes* species mosquito. In endemic regions with high circulation of ZIKV and multiple DENV serotypes, and the presence of cross-reactive or sub-neutralizing antibodies that could enhance dengue disease severity, alternative approaches to antibody-induced vaccination must be considered. This work describes a novel strategy for the

repository, within the manuscript itself or uploaded as supplementary information.

**Funding:** Research reported in this publication was supported by the Labex IBEID (ANR-10-LABX-62-IBEID to A.S.), the National Institute of Allergy and Infectious Diseases of the National Institutes of Health under Award Number U01AI151758 to A.S., and Southeast Asia – Europe Joint Funding Scheme for Research and Innovation (grant ANR-17-ASIE-0008 Programme ASIE 2017 to A.S. and T.C. The content is solely the responsibility of the authors and does not necessarily represent the official views of the National Institutes of Health. The funder played no role in study design, data collection and analysis, decision to publish, or preparation of the manuscript. The content is solely the responsibility of the authors and does not necessarily represent the official views of the National Institutes of Health.

development of a T-cell based vaccine composed of immunodominant T cell epitopes from ZIKV. This vaccine induces a significant protection against ZIKV infection in mice expressing different HLA class I molecules, in the absence of neutralizing antibodies. Likewise, DNA vaccination with mosaic sequence enriched in DENV T cell epitopes with identical sequences between DENV1 and DENV2 induces a significant protection against these 2 DENV serotypes, without inducing neutralizing antibodies. This study paves the way for T-cell based vaccines that could overcome the risk for antibody-mediated enhancement and protect against multiple flavivirus infections.

## Introduction

Zika virus (ZIKV) is an enveloped positive-sense, single-stranded RNA virus which encodes 3 structural proteins Env (E), membrane (M) and capsid (C), that form the viral particles and 7 non-structural (NS) proteins NS1, NS2A, NS2B, NS3, NS4A, NS4B, and NS5, involved in the processing, replication, and assembly of new viruses [1]. ZIKV transmission occurs via the bite of an infected *Aedes* mosquito and in some cases through human sexual contact or the maternal fetal route [2]. Since its first identification in 1947 in the Zika forest in Uganda, Zika has caused small sporadic outbreaks in Africa, Asia and South America, and a large epidemic in 2007 on the island of Yap, in Micronesia, where about 75% of the population was infected [3]. Subsequent outbreaks with higher number of cases occurred in 2013–2014 in French Polynesia and in 2015 in South America, and more specifically in Brazil [4–6]. The illness is usually asymptomatic or mild, with symptoms lasting a few days. In a low percentage of cases, neurological complications can occur such as Guillain-Barré syndrome or congenital malformations [7,8]. In 2022, about 40,000 cases were still reported in the Americas, mainly in Brazil (https://www.paho.org/plisa).

ZIKV shares many similarities with other flaviviruses, more specifically with dengue virus (DENV), with the highest sequence identity between the non-structural proteins NS3 and NS5 of the 4 DENV serotypes (DENV1-4) and ZIKV [9,10]. While ZIKV and DENV share 55–58% sequence identity in the E protein, specially between ZIKV and DENV4, sequence identity is 67–68% for the NS3 and NS5 proteins [10]. Strikingly, while the antibody response against DENV and ZIKV is directed mainly against structural proteins such as envelope protein (the main target of neutralizing antibodies), T cells target both non-structural proteins and structural proteins. The immunodominant epitopes vary between DENV and ZIKV infection and between CD4+ and CD8+ T cells [10,11]. Understanding the specific immune response to ZIKV is complicated, given the structural similarities between DENV and ZIKV, the high level of DENV seroprevalence in areas where ZIKV is circulating and the presence of cross-reactive antibodies and/or T cells to DENV in ZIKV-infected individuals [12,13].

Studies in mice and non-human primates have shown that, in the absence of pre-existing immunity to DENV, a primary ZIKV infection induces not only ZIKV-specific antibodies, but also antibodies that cross-react to DENV and that can mediate enhancement of DENV2 infection [14–16]. Further, human epidemiological and clinical studies confirmed a greater risk of symptomatic and severe DENV2 disease in children with prior ZIKV infection [17]. The risk to develop a symptomatic and severe DENV infection, especially DENV2, depends on the quality of neutralizing antibodies and the level of pre-existing anti-DENV or anti-ZIKV antibodies, with a higher risk associated with lower levels of pre-existing antibodies to DENV or ZIKV [18,19].

Conversely, in areas with major dengue epidemics over 6 years ago, an increased risk of microcephaly was observed after ZIKV infection [20]. Infected children produced cross-

reactive and poorly neutralizing antibodies that can mediate Antibody Dependent Enhancement (ADE) [18]. In contrast, in a large cohort study in Brazil, the presence of high pre-existing DENV antibody titers was shown to be associated with a reduced risk of ZIKV infection or symptoms [21]. While these studies clearly demonstrate a strong correlation between a high antibody titer and protection against disease severity, they also raise the concern that ZIKV-specific antibodies, induced following natural ZIKV infection or vaccination, could enhance dengue disease severity. In this respect, alternative approaches to the development of DENV and ZIKV vaccines that minimize the generation of cross-reactive antibodies are worth considering [22,23].

CD8+ T cells have been demonstrated to play a protective role during primary ZIKV infection, similar as what has been observed during DENV infection [24]. In humans, ZIKV-specific CD8+ T cells are characterized by a polyfunctional IFN-γ signature [25]. In mice lacking the type I interferon receptor in myeloid cells (LysMCre$^+$IFNAR$^{fl/fl}$ mice), depletion of CD8 + T cells resulted in impaired ZIKV clearance, whereas adoptive transfer of ZIKV-immune CD8+ T cells reduced viral burdens [26]. Even though the infiltration of CD8+ T cells into the central nervous system during primary ZIKV infection has been associated with neuropathology in *ifnar*$^{-/-}$ mice [27], when transferred in an appropriate number and differentiation state, these CD8+ T cells were shown to be essential in the control of viral replication in brain [28]. Together, these observations argue in favor of a protective role of ZIKV-specific CD8+ T cells against ZIKV infection. Likewise, the role of CD4+ T cells in protecting against ZIKV infection and disease has also been demonstrated, although results vary according to the mouse model used, the age of infected animals, the route of virus inoculation, as well as the different tissues and organs targeted by these effector T cells [29–31]. In interferon α/β receptor-deficient HLA-DRB1*0101 transgenic mice, dengue and Zika virus cross-reactive CD4 Th1 cells were shown to be protective against ZIKV infection, in an antibody-independent manner [32]. In rhesus macaques, depletion of CD4 T cells before DENV or ZIKV infection resulted in a significant increase in viremia, and an altered quality of the humoral immune response [33]. Therefore, CD4+ T cells are necessary to guarantee an optimal and type-specific humoral immune response against primary DENV infection but also against heterologous sequential DENV/ZIKV infections [33].

The CD4 T cell response is skewed to a Th1 response and is directed against epitopes located in E, NS1, NS3, NS4B and NS5 [22,30]. In non-DENV exposed individuals, ZIKV-specific CD4 and CD8 T cells target both structural (E, prM and C) and non-structural proteins. In contrast, in DENV-exposed individuals T cells are mainly directed towards non-structural proteins such as NS3, NS4B and NS5 that contain peptides with identical amino acid sequences between DENV and ZIKV [22,23,34], a result that has important implications for DENV and ZIKV T cell vaccine development. In secondary DENV infection, it has been proposed that T cells primed after the primary DENV infection could hamper the development of the T cell response towards the secondary infecting serotype, in an original antigenic sin scenario [35]. Evidence for a role of such cross-reactive T cells in the pathogenesis of a secondary ZIKV infection is lacking.

Approved and most advanced DENV vaccines use tetravalent live attenuated viruses, with the major aim of inducing broad cross-serotype neutralizing antibodies. However, concerns remain on vaccine-mediated enhancement of disease due to waning of cross-reactive antibodies. A strategy that circumvents this complexity is the development of T-cell vaccines containing immunodominant T cell epitopes. Vaccination is aimed to elicit CD4 and CD8 T cell responses, without inducing potential enhancing antibodies [32,36,37]. With this in mind, numerous immunodominant T cell epitopes have been identified from the ZIKV proteome,

either in WT mice, in type I IFN receptor deficient mice [37,38], in HLA class I or class II transgenic mice [39,40], or in humans [22,41–43].

In this study, we designed a mosaic sequence called ZIKV-NS poly-epitope, enriched in immunodominant T cell epitopes located in the C, NS1, NS3, NS4B and NS5 proteins. We utilized a DNA vaccine platform using the tetrafunctional amphiphilic block copolymer 704 that has been shown to promote protective immunity against various cancers and viral infections [44–47]. We show that immunization of HLA-A*2402 and -B*0702 transgenic mice with a DNA encoding ZIKV-NS polyepitope formulated with 704 elicits a significant T cell response against numerous T cell epitopes and protection against ZIKV infection, in the absence of neutralizing or enhancing antibodies. We also show that ZIKV-NS poly-epitope, which contains a limited number of antigenic sequences showing cross-reactivity with DENV2, does not induce a significant protection against DENV2. In contrast, a DENV1-NS poly-epitope that contains a higher number of epitopes with identical sequences between DENV1 and DENV2 induces a significant protection against both DENV1 and DENV2 infection in the absence of humoral immunity. These data strongly support the usage of poly-epitope vaccines consisting of multiple antigenic T cell epitopes, to induce T cell-based immune protection in the absence of neutralizing or sub-neutralizing antibodies.

## Methods

### Ethics statement

All animal experiments were performed in accordance with the recommendations of French Ministry of Higher Education and Research and approved by the ethics committee in animal experimentation (Comité d'éthique en experimentation animale, CETEA n˚89) under reference APAFIS#321814–2021052710286548 v1.

### Nucleic acids preparation and formulation

ZIKV-NS and DENV1-NS polyepitopes (patent No 17306553.3, initially filled on November 9, 2017 and patent WO2015/1957565, initially filled on June 23, 2014, respectively) were cloned into pVAX1 plasmid (ThermoFischer Scientific, Les Ulis, France). The antigenic sequences from DENV1-NS and ZIKV-NS poly-epitopes are presented in supplementary information section (S1 Table). Control plasmid consisted in pVAX1 without transgene in the open reading frame. All plasmids prepared were purified using EndoFree plasmid purification columns (Qiagen, Courtaboeuf, France) and were confirmed to be free of endotoxin contamination (endotoxin 4,44 EU/mg plasmid DNA). The tetrafunctional block copolymer 704 was kindly supplied by In-cell-Art (Nantes, France). Plasmid DNA was formulated with 704 immediately prior to *i.m.* injection as previously described [46].

### Mouse vaccination

HLA-A*2402, HLA-B*07:02 and HLA-B*07:02/IFNAR1 transgenic mice 8 weeks of age were housed under conventional conditions according to Institut Pasteur guidelines. For intramuscular DNA vaccination, mice were anesthetized with isoflurane (1–3% for maintenance, up to 5% for induction) in oxygen from a precision vaporizer (Tem Sega, Pessac, France), then different DNA/704 formulations were injected at D0 and D21 into both *tibialis anterior* muscles using an Insumed Pic Indolore 30G syringe (Artsana, Grandate,Italy). In all cases, the injection volume was 50 µl per injection site. After vaccination, spleen samples were collected two weeks after the last immunization to analyze the cellular immune response.

## IFN-γ and Granzyme B ELISPOT Assays

Class I restricted IFNγ secretion was determined by ELISpot (Diaclone, Besançon, France), as a marker for the presence of ZIKV or DENV specific CTL. The mixture of 15-mer overlapping peptides covering the whole poly-epitope sequence was used as representative ZIKV or DENV T cell epitopes. Live splenocytes were counted on a hemocytometer slide by Trypan blue exclusion, resuspended at $1\times10^6$/mL in complete medium (α-MEM supplemented with 10% HyClone Fetal Bovine serum, 2mM L glutamine, 5 units/mL penicillin, and 5 μg/mL streptomycin (1% PS) Hepes, Pyruvate, Non-Essential Amino Acids (NEAA) and β-mercaptoethanol ($5\times10^{-5}$M), then distributed in triplicate at $4\times10^5$ cells/well. Cells were incubated overnight at 37˚C and 5% CO2 in the presence of 5 μg/mL Concanavalin A, or 4 μg/mL peptide pool. After incubation, the wells were washed three times with PBS-0.05% Tween 20, and then incubated with 50 μl biotinylated anti-mouse IFN-γ (BD bioscience, France) at 1 μg/ml in PBS Tween for 1h at room temperature. The spots were developed using streptavidin-alkaline phosphatase (Mabtech, Stockholm, Sweden) and BCIP/NTB substrate (Promega, Madison, MI, USA) then counted using an automated ELI-SPOT reader (Immunospot, Cellular Technology Limited, Cleveland, OH, USA). For Granzyme B ELISpot assay, the mouse Granzyme B ELISpot detection kit was used (R&D systems).

## Mouse challenge

Two weeks after final vaccination, mice received one intraperitoneal injection of 2 mg anti-IFNAR antibody (MAR1-5A3) and the day after, intraperitoneal inoculation of $10^3$ pfu zika virus from French Guyana (FG-15G strain) or systematically by retro-orbital injection of $10^6$ pfu Dengue serotype 2 virus (A0824528 IPC strain from Phillippe Dussart, Institut Pasteur, Phnom Penh, Cambodia).

## Quantification of viral loads

For assessment of viral replication in infected mice, viral RNA was extracted from mouse serum using the QIAamp Viral RNA kit (Qiagen, Hilden, Germany) according to the manufacturer's instructions. Synthesis of cDNA was achieved using the MMLV cDNA synthesis kit (Invitrogen). Viral cDNA was quantified using the Applied Biosystems Master Mix TaqMan Universal II on a QuantStudio 12K Flex Real-time PCR system (Life Technologies) using standard cycling conditions. Quantification of cDNA was performed using Ct values of samples. The following primer sets were used:

Zikv Probe: 5'/56-FAM/ AGCCTACCTTGACAAGCAATCAGAGACTCAA /3MGBEc/-3'
Forward: 5' CCGCTGCCCAACACAAG 3'
Reverse: 5' CCACTAACGTTCTTTTGCAGACAT 3'
Denv-2 Probe: 5'/56-FAM/ CTCDCGCGTTTCAGCATATTG /3MGBEc/-3'
Forward: 5'GCAGATYTCTGAAYAACCAACG 3'
Reverse: 5' AGCATTCCAAGTGAGAATCTCT 3'

## DENV-specific anti-NS3 antibodies

NS3 protein (GeneScript) was adjusted to 5μg/mL with bicarbonate/carbonate buffer (100 mM, pH 9.2). 96-well plates (Maxisorb, Greiner Bio-One, Frickenhausen, Germany) were covered with 100ul of NS3 protein, sealed and incubated at 4˚C overnight. The coated plates were then washed 5 times with washing buffer (PBS, 0,05% Tween 20) and blocked with blocking buffer (PBS, 0.05% tween 20, 3% low-fat milk) for 1 hour at 37˚C. Mice serum was diluted 1:100 in the blocking buffer and 50μl of diluted sera was added. After 1 hour, the plate was washed 5 times with washing buffer and 50 μl of Goat Anti-Mouse IgG antibody conjugated

with HRP (GeneTex) (dilution 1:2000) was added. After 1 hour incubation, the plate was washed 5 times and 50 µl of TMB substrate (BioLegend) was added. The color reaction was stopped after 15 minutes by adding 100 µl of 0.18M $H_2SO_4$ per well. The intensity of the color reaction was measured at 450 nm and reported as optical density (OD). OD of empty vector immunized mice was used to calculate the Cutoff for positivity based on the following formula: Cutoff = mean + 2 standard deviation.

## Immunophenotyping of T cells

Spleen cells from immunized mice were stained with FITC-conjugated anti-CD3, V500-conjugated anti-CD4, PerCP-conjugated anti-CD8, PE-Vio-conjugated anti-KLRG1 and APC-conjugated anti-CD62L antibodies from eBiosciences. Flow cytometry analyses were performed on a Myltenyi MACSQuant analyser.

## Flow cytometry-based neutralization assay

Vero cells (ATCC No. CCL-81) were cultured in DMEM (Dulbecco's Modified Eagle Medium) supplemented with 5% FBS (Gibco), 100 U/ml penicillin, 100 µg/ml streptomycin (Gibco) and seeded into 96 well culture plates one day prior. Heat inactivated serum from mice was serially diluted starting from 1:200 in RPMI and incubated with DENV2 (New Guinea) or ZIKV (NC) corresponding to MOI of 1 for 1 hour at 37˚C, 5% $CO_2$. Cells infected with DENV2 or ZIKV in the absence of sera were used as control. Immune complexes were transferred to 96-well culture plates containing the cells and were incubated for 90 minutes at 37˚C, 5% $CO_2$. After incubation, the plates were washed and DMEM complete medium was added. Cells were incubated at 37˚C, 5% $CO_2$. After 24 hours, the cells were detached by trypsin 2X and stained with Zombie Aqua viability dye (BioLegend) to exclude dead cells. Then the cells were fixed, permeabilized and stained for the presence of DENV using anti-DENV E protein antibody (clone 4G2, ATCC HB 112) labelled with Alexa Fluor 488 (Invitrogen) and analysed by flow cytometry (FACSCanto II, BD Biosience). Antibody neutralization was plotted as percentage of inhibition of infection.

## Antibody-dependent enhancement (ADE) assay

Human myelomonocyte cell lines U937 (ATCC CRL 1593.2) were cultured in RPMI (Gibco) supplemented with 10% FBS (Gibco), 100 U/ml penicillin, 100 µg/ml streptomycin (Gibco) and 1% L glutamine (Gibco). Heat inactivated serum from mice was diluted 1:400 in RPMI with 2% FBS and incubated with DENV2 (New Guinea) or 1:1600 incubated with ZIKV (NC) corresponding to MOI of 1 for one hour at 37˚C, 5% $CO_2$ to form immune complexes. Direct infection of U937 cells with DENV2 or ZIKV in the absence of plasma was used as control. Immune complexes were transferred to 96 well round-bottom plates containing 80,000 U937 cells/well. The plates were incubated for 90 minutes at 37˚C, 5% $CO_2$. After infection, cells were washed and incubated for 72 hours at 37˚C, 5% $CO_2$. After incubation, the cells were stained with Zombie Aqua viability dye (BioLegend) to exclude dead cells. Then, the cells were fixed, permeabilized and stained for the presence of DENV and ZIKV using anti-DENV E protein antibody (clone 4G2, ATCC HB 112) labelled with Alexa Fluor 488 (Invitrogen) and analysed by flow cytometry (FACSCanto II, BD Biosience). ADE was plotted as percentage of infected cells.

## Antibody dependent cellular cytotoxicity (ADCC) assay

Mouse ADCC Reporter Bioassay was performed according to manufacturer's instructions (Promega). Briefly, 25µl of 1:25 diluted heat inactivated serum from mice was added to 25µl of

25 × 103 A549 cells (ATCC CCL-185) either infected with DENV2 (MOI2) or mock infected and 25ul of $75 \times 10^3$ Jurkat reporter cells (1:3 Target:Effector ratio). For the control, 1 μl of 3.56 mg/ml anti-CD20 antibodies were added to 25ul of 25x103 Raji cells (ATCC CCL-86) and 25ul of Jurkat reporter cells (1:3 Target:Effector ratio). The cells were then incubated at 37˚C and 5% $CO2$ in a cell culture incubator for 6 hours. At the end of the incubation time, 75μl of Bio-Glo Assay Reagent (Promega) were added into each well and the plates were read after 15 min incubation using the Cytation 5 image reader (BioTek). Assay results are reported as relative luminescence unit values (RLUs).

## Results

### Design of a ZIKV mosaic antigen enriched in T cell epitopes

To evaluate the feasibility of a T cell DNA delivered by 704 to induce immune responses, we designed a mosaic sequence encoding dominant T cell epitopes of ZIKV. Using human blood samples from ZIKV-infected donors, numerous CD4 and CD8 T cell epitopes were identified, located in the Capsid, the NS1, NS3, NS4B and NS5 proteins [22]. A mosaic sequence encoding these dominant epitopes has been synthetized, the ZIKV-NS poly-epitope (patent EP 17 306553.3, filed on November 9, 2017). This poly-epitope is composed of 1 region located in the Capsid, 3 distinct regions in the NS1, 2 regions in the NS3, 1 region in the NS4B and 4 regions in the NS5 proteins (Table 1). Among the epitopes selected in the ZIKV-NS poly-epitope, 19 peptides reveal 80% or more sequence identity with peptides derived from the two dengue virus serotypes, DENV1 and DENV2. Interestingly, the strongest T cell responses and the highest frequency of responders were observed against the NS5.1 $_{2817-2831}$, the NS5.1 $_{2865-2879}$, the NS5.1 $_{2985-2999}$, and the NS5.1 $_{3001-3015}$ peptides located in the NS5.1 region of ZIKV, and against the NS5.2 $_{3093-3107}$ peptide located in the NS5.2 region of ZIKV. Notably, these regions are also enriched in peptides with similar or identical sequences between ZIKV, DENV1 and DENV2 [22]. Taken together, this suggests that in humans, T cells raised against peptides from ZIKV sequence are potentially cross-reactive to peptides derived from DENV1 or DENV2.

### Immunogenicity of the ZIKV-NS poly-epitope

To determine whether the ZIKV-NS poly-epitope can be produced endogenously and processed into HLA class I-restricted peptides, HLA-A*2402 and -B*0702 HLA class I transgenic mice were immunized in a prime/boost regimen with 704/DNA encoding ZIKV-NS poly-epitope. T cell responses were tested by IFNγ-ELISpot against 15-mer overlapping peptides covering the whole poly-epitope sequence. Mice expressing HLA- A*2402 or -B*0702 were chosen as these alleles are frequently expressed, are associated with low and high frequency and magnitude of T cell response to DENV in humans [48]. Moreover, using the data available in the Immune Epitope Database (IEDB:: https://www.iedb.org (assessed on 5 December 2023)), we observed that the ZIKV-NS poly-epitope contains a significant number of T cell epitopes with a strong potential for binding to these two alleles. As a control of 704-based DNA immunization, these HLA class I transgenic mice were also immunized with the DENV1-NS poly-epitope, previously shown to induce a strong CD8 T cell response to several peptides restricted by the HLA-A*0201, -A*2402, -B*0702 and -B*3501 class I molecules [49]. The frequency of splenocytes obtained from HLA-A*2402 and -B*2702 transgenic mice immunized with 704/DNA encoding ZIKV-NS and DENV1-NS and producing IFN-γ in response to peptide stimulation is shown in Fig 1.

**Immunogenicity in HLA-A*2402 transgenic mice.** HLA-A*2402 transgenic mice were immunized with 704 complexed ZIKV-NS poly-epitope or DENV1-NS poly-epitope (Fig 1A, 1B, 1C and 1D), and T cell epitopes were identified by IFN-γ ELISpot assay, using 15-mer

**Table 1. Immunodominant epitopes of the ZIKV proteome selected in the ZIKV-NS poly-epitope.**

| Region of ZIKV | Sequence | Frequency of responders (%) | Total no. of SFCs | Avg. no. of SFCs (positive donors) | Denv1 sequence identity (%) | Denv2 sequence identity (%) |
|---|---|---|---|---|---|---|
| Capsid $_{9-23}$ | GGFRIVNMLKRGVAR | 10 | 245 | 122 | 40 | 40 |
| Capsid $_{21-35}$ | VARVSPFGGLKRLPA | 10 | 80 | 40 | 26.6 | 26.6 |
| Capsid $_{29-43}$ | GLKRLPAGLLLGHGP | 15 | 435 | 145 | 46.6 | 46.6 |
| Capsid $_{49-63}$ | AILAFLRFTAIKPSL | 15 | 425 | 140 | 60 | 53.3 |
| Capsid $_{77-91}$ | AMEIIKKFKKDLAAM | 15 | 310 | 103 | 33.3 | 26.6 |
| Capsid $_{85-99}$ | KKDLAAMLRIINARK | 15 | 395 | 131 | 53 | 40 |
| NS1.1 $_{813-827}$ | VFVYNDVEAWRDRYK | 25 | 575 | 115 | 40 | 33.3 |
| NS1.1 $_{833-847}$ | PRRLAAAVKQAWEDG | 15 | 325 | 108 | 60 | 46.6 |
| NS1.1 $_{857-871}$ | MENIMWRSVEGELNA | 10 | 310 | 155 | 53.3 | 46.6 |
| NS1.1 $_{862-870}$ | WRSVEGELN | 10 | 230 | 115 | 44 | 44 |
| NS1.1 $_{884-892}$ | VGSVKNPMW | 10 | 220 | 110 | 33.3 | 22 |
| NS1.1 $_{885-899}$ | GSVKNPMWRGPQRLP | 25 | 440 | 88 | 20 | 20 |
| NS1.1 $_{905-919}$ | LPHGWKAWGKSYFVR | 15 | 325 | 108 | 33.3 | 33.3 |
| NS1.2 $_{941-955}$ | HRAWNSFLVEDHGFG | 15 | 485 | 161 | 66,6 | 73 |
| NS1.2 $_{958-966}$ | HTSVWLKVR | 15 | 227 | 75 | 55.5 | 55.5 |
| NS1.2 $_{962-970}$ | WLKVREDYS | 15 | 325 | 108 | 55.5 | 44 |
| **NS1.2 $_{989-1003}$** | **HSDLGYWIESEKNDT** | **10** | **615** | **307** | **80** | **73.3** |
| NS1.3 $_{1069-1083}$ | IRFEECPGTKVHVEE | 10 | 380 | 190 | 46.6 | 13.3 |
| NS1.3 $_{1074-1082}$ | CPGTKVHVE | 15 | 360 | 120 | 55.5 | 11.1 |
| NS3.1 $_{1633-1647}$ | PAGTSGSPILDKCGR | 25 | 900 | 180 | 53.3 | 66.6 |
| NS3.1 $_{1638-1646}$ | GSPILDKCG | 15 | 530 | 176 | 55.5 | 77.7 |
| NS3.1 $_{1644-1652}$ | KCGRVIGLY | 15 | 565 | 188 | 55.5 | 66.6 |
| NS3.1 $_{1645-1659}$ | CGRVIGLYGNGVVIK | 25 | 990 | 198 | 60 | 66.6 |
| **NS3.1 $_{1650-1658}$** | **GLYGNGVVI** | **10** | **360** | **180** | **88.8** | **88.8** |
| **NS3.2 $_{1701-1715}$** | **GKTRRVLPEIVREAI** | **5** | **105** | **105** | **86.6** | **80** |
| **NS3.2 $_{1725-1739}$** | **APTRVVAAEMEEALR** | **10** | **205** | **102** | **80** | **100** |
| NS3.2 $_{1749-1763}$ | AVNVTHSGTEIVDLM | 10 | 300 | 150 | 66.6 | 60 |
| **NS3.2 $_{1785-1799}$** | **IMDEAHFTDPSSIAA** | **10** | **425** | **212** | **93.3** | **93.3** |
| **NS3.2 $_{1809-1823}$** | **MGEAAAIFMTATPPG** | **10** | **375** | **187** | **100** | **93.3** |
| **NS3.2 $_{1813-1827}$** | **AAIFMTATPPGTRDA** | **10** | **470** | **235** | **80** | **80** |
| **NS3.2 $_{1816-1824}$** | **FMTATPPGT** | **10** | **220** | **110** | **88.8** | **88.8** |
| NS3.2 $_{1825-1839}$ | RDAFPDSNSPIMDTE | 10 | 310 | 155 | 53.3 | 66.6 |
| NS4B $_{2373-2387}$ | LTPLTLIVAIILLVA | 10 | 195 | 97 | 60 | 53.3 |

*(Continued)*

**Table 1.** (Continued)

| Region of ZIKV | Sequence | Frequency of responders (%) | Total no. of SFCs | Avg. no. of SFCs (positive donors) | Denv1 sequence identity (%) | Denv2 sequence identity (%) |
|---|---|---|---|---|---|---|
| NS4B 2381–2389 | AIILLVAHY | 15 | 95 | 31 | 66.6 | 77.7 |
| NS5.1 2813–2827 | WFFDENHPYRTWAYH | 5 | 1580 | 1240 | 66.6 | 66.6 |
| NS5.1 2816–1824 | DENHPYRTW | 10 | 290 | 145 | 66.6 | 66.6 |
| **NS5.1 2817–2831** | **ENHPYRTWAYHGSYE** | **10** | **1385** | **692** | **80** | **80** |
| NS5.1 2829–2843 | SYEAPTQGSASSLIN | 15 | 480 | 160 | 60 | 60 |
| **NS5.1 2845–2859** | **VVRLLSKPWDVVTGV** | **15** | **495** | **165** | **73.3** | **80** |
| **NS5.1 2865–2879** | **TDTTPYGQQRVFKEK** | **15** | **5840** | **1946** | **93.3** | **93.3** |
| **NS5.1 2868–2876** | **TPYGQQRVF** | **15** | **2160** | **720** | **88.8** | **88.8** |
| NS5.1 2893–2907 | QVMSMVSSWLWKELG | 20 | 595 | 148 | 40 | 53.3 |
| NS5.1 2945–2953 | EAVNDPRFW | 15 | 570 | 190 | 77.7 | 77.7 |
| NS5.1 2948–2956 | NDPRFWALV | 10 | 260 | 130 | 66.6 | 66.6 |
| NS5.1 2949–2957 | DPRFWALVD | 10 | 335 | 167 | 66.6 | 77.7 |
| **NS5.1 2985–2999** | **EFGKAKGSRAIWYMW** | **20** | **1085** | **271** | **100** | **100** |
| **NS5.1 3001–3015** | **GARFLEFEALGFLNE** | **20** | **2905** | **726** | **93.3** | **100** |
| NS5.2 3065–3079 | RFDLENEALITNQME | 15 | 515 | 171 | 60 | 53.3 |
| NS5.2 3069–3083 | ENEALITNQMEKGHR | 15 | 375 | 125 | 53.3 | 46.6 |
| **NS5.2 3085–3099** | **LALAIIKYTYQNKVV** | **15** | **555** | **185** | **73.3** | **80** |
| **NS5.2 3089–3103** | **IIKYTYQNKVVKVLR** | **15** | **500** | **166** | **80** | **73.3** |
| **NS5.2 3093–3107** | **TYQNKVVKVLRPAEK** | **15** | **615** | **205** | **80** | **73.3** |
| NS5.3 3145–3159 | MEAEEVLEMQDLWLL | 20 | 455 | 113 | 26.6 | 26.6 |
| NS5.4 3217–3231 | KPSTGWDNWEEVPFC | 15 | 355 | 118 | 60 | 60 |
| NS5.4 3257–3265 | IGRARVSPG | 5 | 195 | 175 | 77.7 | 77.7 |
| **NS5.4 3259–3267** | **RARVSPGAG** | **10** | **180** | **90** | **88.8** | **77.7** |
| **NS5.4 3322–3330** | **KGEWMTTED** | **10** | **185** | **92** | **66.6** | **88.8** |

Amino acid sequences in bold have at least 80% sequence identity with DENV1 or DENV2

peptides overlapping by 11 amino acids. All the antigenic peptides identified from HLA-A*2402 mice immunized with the ZIKV-NS poly-epitope correspond to epitopes previously identified in the human [22,43,50], which include 6 epitopes in the NS3.1 and NS3.2 regions (Fig 1A), and 6 and 4 epitopes in the NS4B and NS5.1 regions, respectively (Fig 1B and 1C). In the NS3.2 region of ZIKV-NS poly-epitope, 3 peptides ZV $_{1790}$, ZV $_{1794}$ and ZV $_{1798}$ reveal at least 80% sequence identity with the NS3 $_{1763-1777}$ and NS3 $_{1762-1776}$ peptides from DENV1 or DENV2, respectively (S1 Table). In the NS4B, among the 6 epitopes identified inducing a substantial T cell response, and previously identified in the human, none of them revealed a significant sequence identity with DENV1 or DENV2 (Tables 1 and S1). In the NS5.1, among the 4 antigenic peptides identified in the HLA-A*2402 transgenic mice (ZV $_{2805-2819}$, ZV $_{2809-2823}$, ZV $_{2829-2843}$ and ZV $_{2833-2847}$, Fig 1C), none of them revealed any significant sequence homology with DENV1 or DENV2 sequences (Tables 1 and S1).

Likewise, in the HLA-A*2402 transgenic mice immunized with 704/DNA encoding DENV1-NS sequence, 5 epitopes were identified, including 1 in the NS3.1 (DV $_{1757-1771}$), 1 in the NS3.2 (DV $_{2032-2040}$) and 3 in the NS5 (DV $_{2765-2779}$, DV $_{2769-2783}$ and DV $_{2793-2807}$) (Fig 1D and S1 Table). Among these epitopes, 2 peptides, DV $_{1757-1771}$ and DV $_{2793-2807}$, revealed 100% and 80% sequence identity with DENV2, respectively. A third peptide (DV $_{2032-2040}$), previously identified as the target of cross-reactive T cells, revealed 78% sequence identity with DENV2 [49] (S1 Table).

**Immunogenicity in HLA-B*0702 transgenic mice.**   In the HLA-B*0702 transgenic mice, 704/DNA immunization with the ZIKV-NS polyepitope induced a significant T cell response to 5 peptides located in the Capsid (ZV $_{20-34}$), the NS3.2 (ZV $_{1702-1716}$ and ZV $_{1722-1736}$) and the NS5.1 (ZV $_{2829-2843}$ and ZV $_{2905-2919}$) regions (Fig 1E). All these 5 epitopes correspond to epitopes previously identified in humans [11,22,50] (and Table 1), including 2 that reveal a strong sequence identity with peptides from DENV2 (the NS3.2 $_{1702-1716}$ from ZIKV has 86.6% sequence identity with the NS3 $_{1774-1788}$ from DENV1 and DENV2 and the NS3.2 $_{1722-1736}$ from ZIKV has 93.3% sequence identity with the NS3 $_{1694-1708}$ from DENV2) (S1 Table).

In these HLA-B*0702 transgenic mice, 704/DNA immunization with DENV1-NS allowed the identification of 8 antigenic peptides, including 2 peptides located in the NS3.1 region (DV $_{1678-1688}$, DV $_{1785-1799}$), 1 peptide in the NS4B (DV $_{2319-2333}$) and 5 peptides in the NS5 region (DV $_{2765-2779}$, DV $_{2769-2783}$, DV $_{2793-2807}$, DV $_{2797-2811}$ and DV $_{2882-2891}$) (Fig 1F). Among these peptides, 6 show 80% or more sequence identity with DENV2 peptides (DV $_{1678-1688}$, DV $_{1785-1799}$, DV $_{2319-2333}$, DV $_{2793-2807}$, DV $_{2797-2811}$ and DV $_{2882-2891}$), of which 3 peptides reveal at least 80% sequence identity with ZIKV peptides (DV $_{1678-1688}$, DV $_{1785-1799}$ and DV $_{2882-2891}$) (S1 Table).A schematic representation of these epitopes within the ZIKV-NS and DENV1-NS poly-epitopes and their potential T cell cross-reactivity is shown Fig 2.

In summary, immunization of HLA-A*2402 transgenic mice with 704/DNA encoding the ZIKV-NS poly-epitope results in the induction of a strong T cell response to 16 distinct peptides, including 3 peptides with a significant sequence identity with DENV2, whereas immunization of HLA-A*2402 with the NS-DENV1 poly-epitope induces a significant T cell response to 5 distinct epitopes, among which 3 peptides have a high sequence identity with DENV2 (Fig 2A and 2B). In addition, immunization of HLA-B*0702 transgenic mice with DNA formulated with 704 encoding the ZIKV-NS sequence induces the activation of T cells against 5 peptides, of which 2 have strong sequence identity with DENV2 (Fig 2A), whereas immunization of HLA-B*0702 mice with DNA encoding the DENV1-NS sequence results in the activation of T cells against 8 peptides, of which 6 have more than 80% sequence identity with DENV2, and 3 have 80% or more sequence identity with ZIKV (Fig 2B).

**Immunogenicity in HLA-B*07:02/IFNAR transgenic mice.**   While in humans, ZIKV inhibits type I IFN signaling by proteasome-mediated degradation of STAT2, it cannot

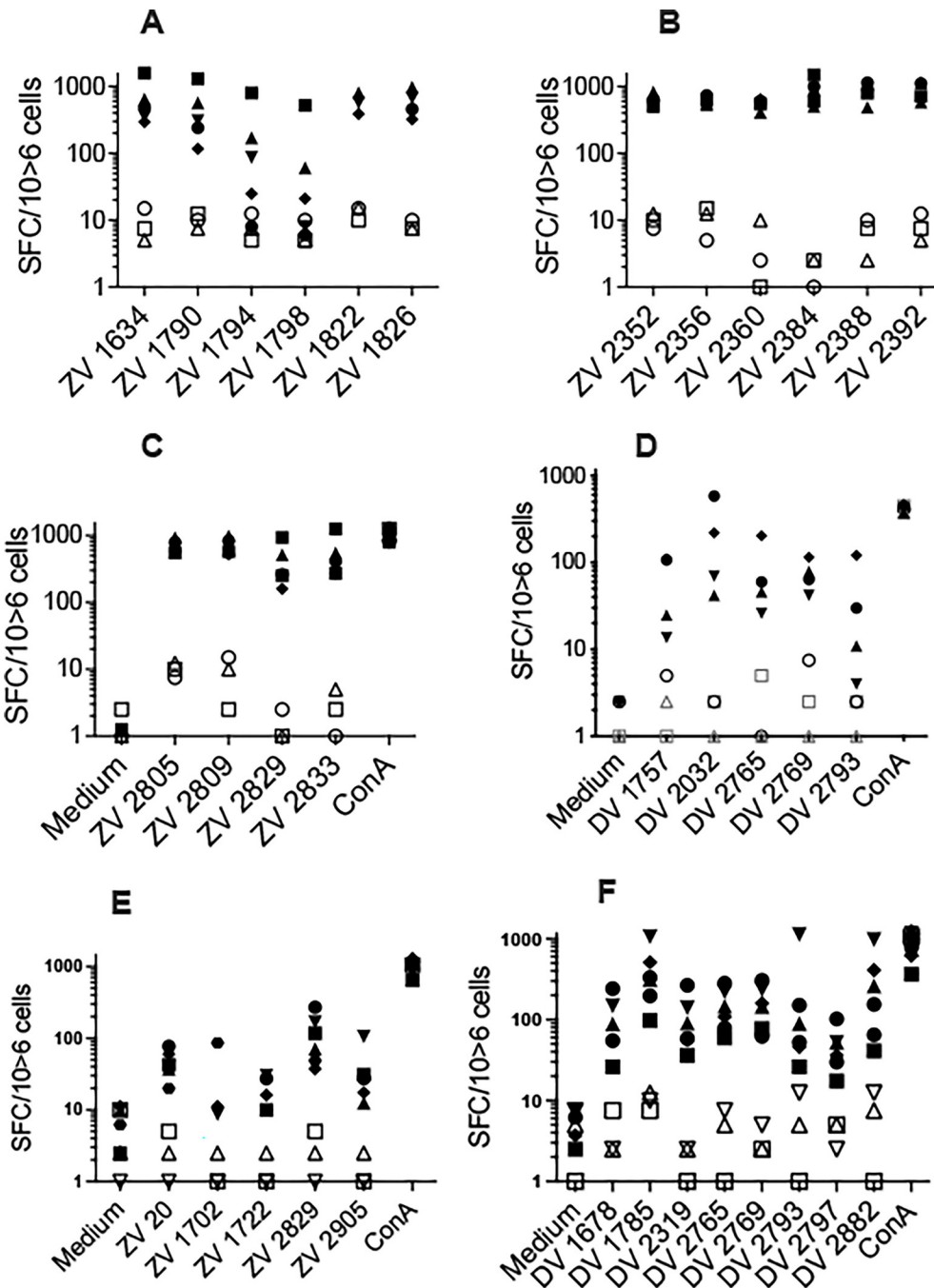

**Fig 1. Identification of immunodominant T cell epitopes in the ZIKV-NS and the DENV1-NS poly-epitopes.**
Three independent experiments were performed in which a total of 5 HLA-A*2402 transgenic mice were immunized at D0 and D21 with 704 complexed ZIKV-NS Poly-epitope (closed symbols) and the empty vector (open symbols), respectively (A, B, C), 4 and 3 HLA-A*2402 transgenic mice received the DENV1-NS Poly-epitope (closed symbols) and the empty vector (open symbols), respectively (D), 6 and 3 HLA-B*07:02 transgenic mice were immunized at D0 and D21 with 704 complexed ZIKV-NS Poly-epitope (closed symbols) and the empty vector (open symbols), respectively (E) and 6 and 3 HLA-B*0702 transgenic mice received the 704 complexed DENV1-NS poly-epitope (closed symbols) and the empty vector (open symbols), respectively (F). Peptides ZV and DV correspond to peptides derived from the ZIK-NS or the DENV1-NS sequences, respectively. Numbers correspond to the position of the first amino acid of the antigenic peptides in the ZIKV or the DENV1 protein sequence. T cell epitopes were identified by IFN-γ ELISpot assay, using 15-mer peptides overlapping by 11 amino acids. ELISpot analysis on dissociated splenocytes cells was performed two weeks after the last immunization. SFC: Spot Forming Cells. Results were expressed as SFC per million of splenocytes.

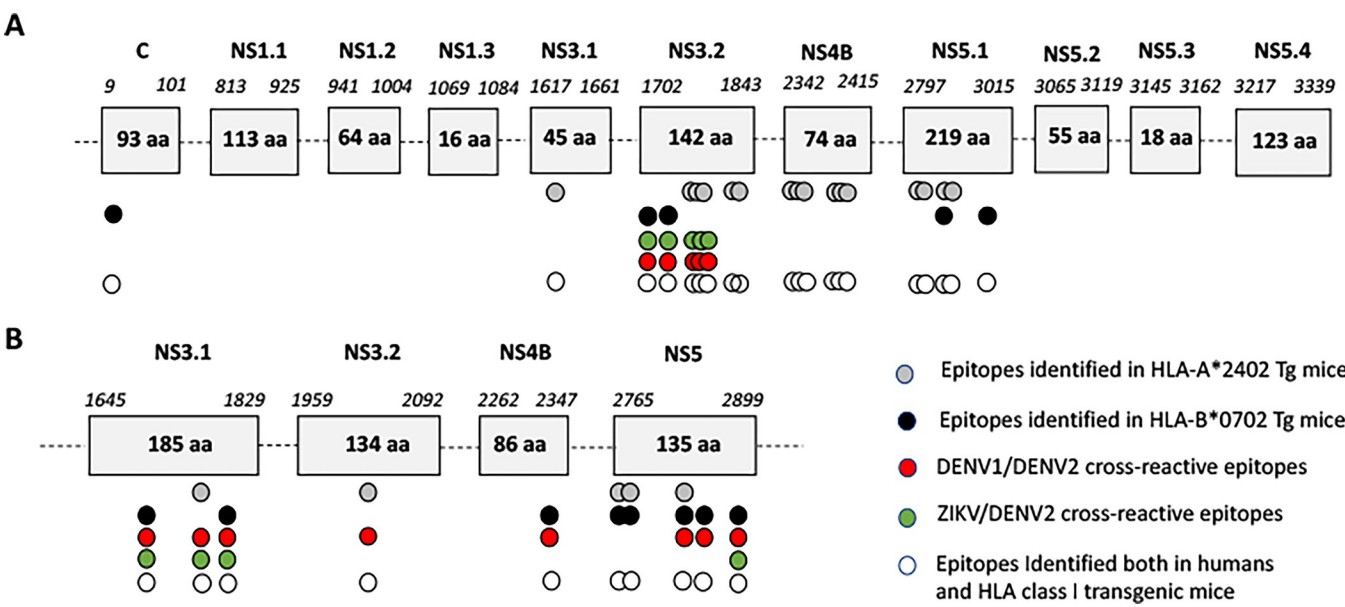

**Fig 2.** Schematic representation of T cell epitopes from the ZIKV-NS poly-epitope (A) or DENV1-NS poly-epitope (B), inducing a significant T cell response in HLA-A*2402 (grey circles) and HLA-B*0702 (black circles) transgenic mice, or in human patients previously infected with dengue or Zika viruses (open circles). Denv1/Denv2 and ZIKV/Denv2 cross-reactive epitopes are represented in red and green circles, respectively. T cell epitopes were identified by IFN-g ELISpot assay, using 15-mer peptides overlapping by 11 amino acids. C, NS1.1, NS1.2, NS1.3, NS3.1, NS3.2, NS4B, NS5.1, NS5.2, NS5.3 and NS5.4 represent the 11 antigenic regions selected in the ZIKV-NS poly-epitope. NS3.1, NS3.2, NS4B and NS5 represent the 4 antigenic regions selected in the DENV1-NS poly-epitope.

antagonize murine IFN-α/β signaling or effector functions, and wild type adult mice are generally resistant to ZIKV and do not develop the typical clinical manifestations of ZIKV infection [51]. Therefore, with the objective to mimic ZIKV pathogenesis in humans and to evaluate candidate vaccines, different mouse models have been developed in which type I IFN signaling is impaired [52,53]. We thus assessed the immunogenicity of 704/DNA encoding DENV1-NS and ZIKV-NS polyepitopes in HLA-B*0702 transgenic mice lacking the IFN-α/β receptor.

In HLA-B*0702 IFN-α/βR$^{-/-}$ mice immunized with 704/DNA encoding DENV1-NS, significant T cell responses were detected against the immunodominant epitopes derived from DENV1-NS poly-epitope. Interestingly, and in agreement with previous studies [52], the magnitude of response was higher in HLA-B*0702 IFN-α/βR$^{-/-}$ mice compared to HLA-B*0702 IFN-α/βR$^{+/+}$ mice. (Fig 3A). A significant and similar T cell response was also observed in HLA-B*0702 IFN-α/βR$^{-/-}$ and HLA-B*0702 IFN-α/βR$^{+/+}$ mice immunized with ZIKV-NS poly-epitope in response to peptides derived from ZIKV. Peptides ZV 1702 and 1706 are an exception, and do not induce any detectable response in the IFN-α/βR$^{+/+}$ mice (Fig 3B), thus confirming the low antigenicity of this epitope (Fig 1E).

## Immunophenotyping of CD8 T cells induced after vaccination with ZIKV-NS and DENV1-NS poly-epitopes

To show that the ZIKV-NS poly-epitope induces the activation of the CD8 T cell compartment, we have also analyzed by flow cytometry the phenotype of CD8+ T cells, in response to the 704 formulated with ZIKV-NS construct or the empty vector as a control. In HLA-A*2402 and -A*0201 transgenic mice, immunization with ZIKV-NS induced a slight increase in HLA-A*2402 mice and a significant increase in CD8 effector memory cells (KLRG1hi CD62Llo) of HLA-A*0201 mice at day 24, but not at day 10 after the secondary immunization

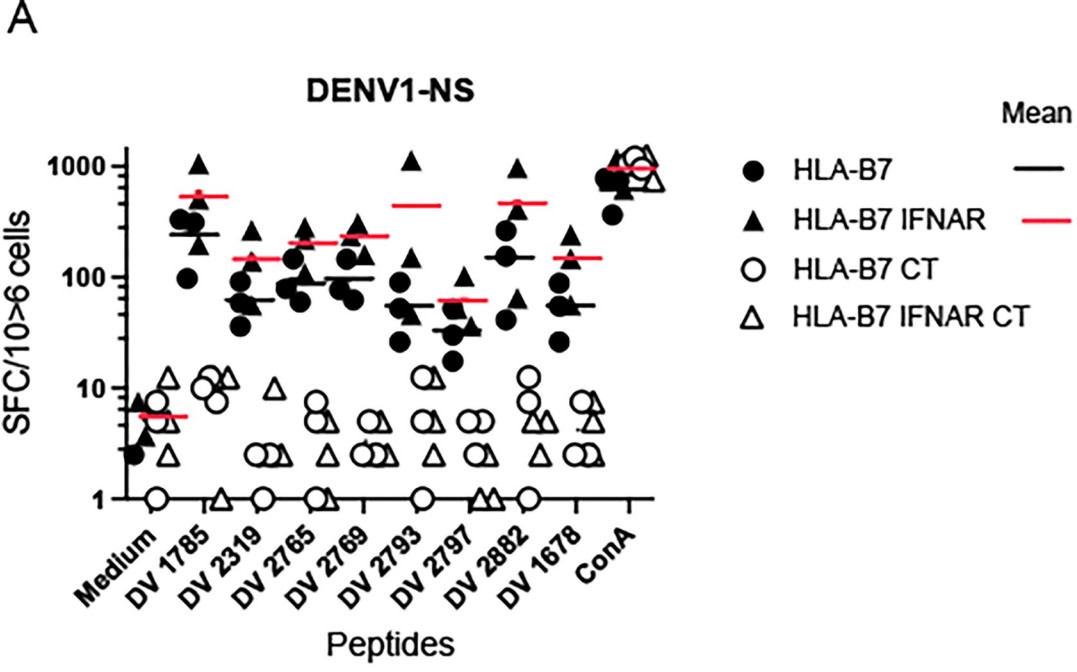

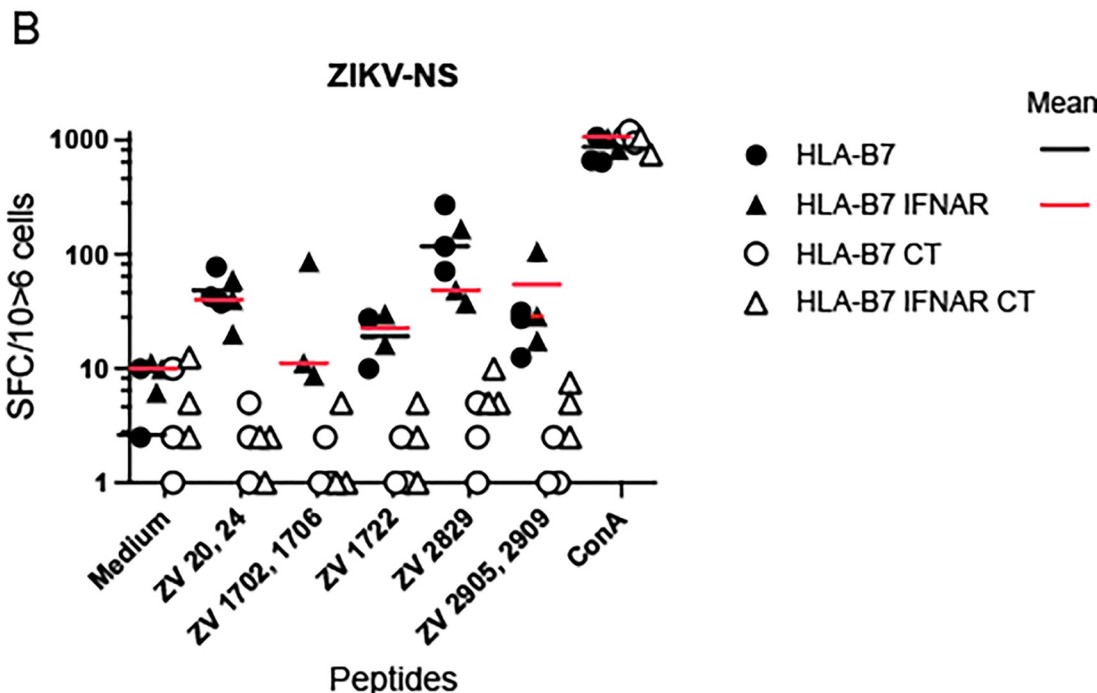

**Fig 3. Identification of immunodominant T cell epitopes in the ZIKV-NS and the DENV1-NS poly-epitopes.** HLA-B*07:02/ IFNAR transgenic mice immunized at D0 and D21 with 704 complexed DENV1-NS Poly-epitope (A) or ZIKV-NS Poly-epitope (B). Three mice per group received the 704 complexed with DENV1-NS or ZIKV-NS (closed symbols) or the empty vector (open symbols) as control. Peptides DV and ZV correspond to peptides derived from the DENV1-NS and ZIKV-NS sequences, respectively. Numbers correspond to the position of the first amino acid of the antigenic peptides in the ZIKV or the DENV1

protein sequence. T cell epitopes were identified by IFN-g ELISpot assay, using 15-mer peptides overlapping by 11 amino acids. ELISpot analysis on dissociated splenocytes cells was performed two weeks after the last immunization. SFC: Spot Forming Cells. Six mice were included per group. Results were expressed as SFC per million of splenocytes.

(Figs 4 and S1). This result confirms that immunization with the ZIKV-NS construct induces the activation of CD8 T cells that bear markers associated with effector memory cells [54]. To determine whether T cells induced after DENV1-NS immunization are functional and can produce cytotoxic enzymes in response to DENV1 peptides, we also determined the frequency of T cells producing IFN-γ and Granzyme B in response to peptide stimulation *in vitro*. As shown in S2 Fig, spleen cells from both the HLA-A*0201 and B*3501 transgenic mice can produce detectable levels of IFN-γ and Granzyme B, in response to antigenic peptides from DENV1. These data strongly support that T cells induced after 704 DENV1 stimulation are functional and are degranulating following stimulation with antigenic peptides.

## Vaccine efficacy of ZIKV-NS and DENV1-NS T cell vaccines in HLA class I transgenic mice

Given the potential of the ZIKV-NS poly-epitope to induce a strong T cell activation against NS epitopes, we wanted to assess vaccine efficacy of the ZIKV-NS poly-epitope in immuno-competent mice expressing HLA-A*2402 or HLA-B*0702 molecules, challenged with live virus. The animals were vaccinated following a prime-boost immunization, with a boost at day 21 after the prime and a challenge at day 36. Since transient blockade of type I IFN signaling was shown to render WT mice susceptible to ZIKV or DENV infection [49,53], without altering T cell responses to antigenic peptides [55], we treated mice with 2mg of the anti-IFNAR antibody MAR1-5A3 one day prior to infection at day 36 with $10^3$ pfu of ZIKV or $10^6$ pfu of DENV2., and quantified viremia from day 1 to 4 after infection.

In the HLA-A*2402 mice, 3 out of 4 mice immunized with the control vaccine developed a significant viremia, at day 2 after the challenge, whereas only 2 out of 5 mice vaccinated with 704/DNA encoding the ZIKV-NS revealed a detectable viremia, which was close to the lower limit of detectable viremia. At day 3 after challenge, all the mice immunized with the control vaccine developed a viremia, whereas no viremia was detected in animals immunized with the ZIKV-NS vaccine (Fig 5A). In the HLA-B*0702 mice, while 5 and 6 out of 6 HLA-B*0702 mice immunized with the control vaccine developed a significant viremia at day 3 and day 4, no viremia was detected in the 5 mice immunized with 704/DNA encoding ZIKV-NS, at all-time points measured (Fig 5B).

Given that DENV1-NS contains 6 epitopes restricted by HLA-B*0702 with almost identical sequences between DENV1 and DENV2 (DV $_{1678}$, DV $_{1785}$, DV $_{2319}$, DV $_{2793}$, DV $_{2797}$ and DV $_{2882}$) (Fig 1F and S1 Table), we aimed to determine the ability of the DENV1-NS to induce a protection against DENV2. In HLA-B*0702 mice immunized with DENV1-NS and challenged with DENV2, viremia was detected in 5 out of 6 mice at day 2 and in all animals at days 3 and 4, after immunization with the control vaccine. In contrast, lower levels of viremia were detected at days 3 and 4 in all animals vaccinated with 704/DNA encoding DENV1-NS (Fig 5C)

We observed that DENV1-NS can activate DENV2 cross-reactive T cells and provide protection against heterotypic infection. Therefore, we aimed to determine whether the ZIKV-NS poly-epitope, which contains several epitopes with nearly identical sequences with DENV2, could also induce a cross-protection against DENV2. We vaccinated HLA-B*0702 mice with 704/DNA encoding the ZIKV-NS poly-epitope. Viremia tended to be lower in the vaccinated mice compared to the control group at days 2 and 3 after the challenge, however, this did not reach significance (Fig 5D).

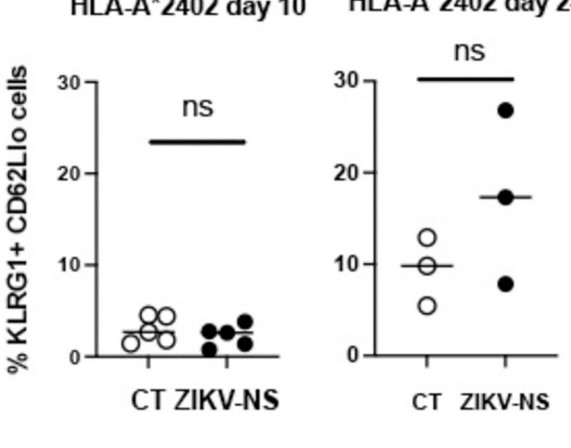

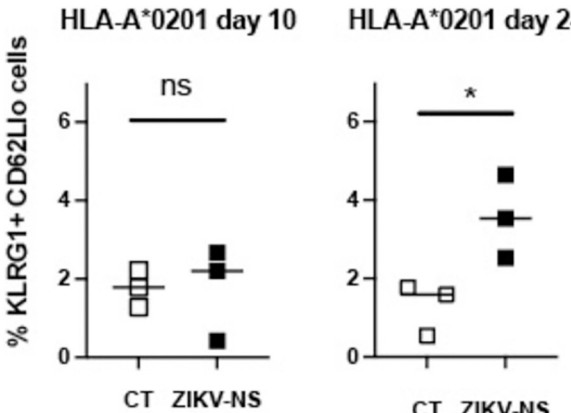

**Fig 4. Expression of markers on CD8+ T cells after immunization of HLA-A*0201 or HLA-A*2402 transgenic mice with 704 complexed ZIKV-NS Poly-epitope (filled symbols) or the empty vector (open symbols).** Three to Five transgenic mice received 2 intramuscular injections of 50 mg of ZIKV-NS or CT vector at day 0 and day 21. Flow cytometry analyses of gated CD3+ CD8+ T cells, for KLRG1 and CD62L expression, were performed 10 days and 24 days after the last immunization. Differences between mice immunized with the ZIKV-NS and the CT vaccines were evaluated using the unpaired t-test (* p<0.05).

Taken together, these results show that immunization of HLA-A*2402 and -B*0702 mice with the ZIKV-NS poly-epitope induces protection against ZIKV infection, but not against DENV2 infection. In contrast, immunization of HLA-B*0702 mice with the DENV1-NS poly-epitope induces serotype cross-reactive protection against DENV2 infection.

## Lack of antibodies induced after DENV1-NS or ZIKV-NS vaccination

To rule out the possibility of antibody-mediated protection against ZIKV or DENV infection of 704/DNA vaccinated animals, we quantified the level of DENV-specific anti-NS3 antibodies, as well as neutralizing antibodies to DENV and ZIKV. As positive control, serum from IFNAR[-/-] mice 35 days after infection with DENV2 or ZIKV was used, whereas serum obtained from animals immunized with the control 704-based vaccine served as negative control. No anti-NS3 IgG was detected in the serum of animals immunized with DENV1-NS or ZIKV-NS [49] (Figs 6A and S3). High levels of neutralizing antibodies to DENV2 were

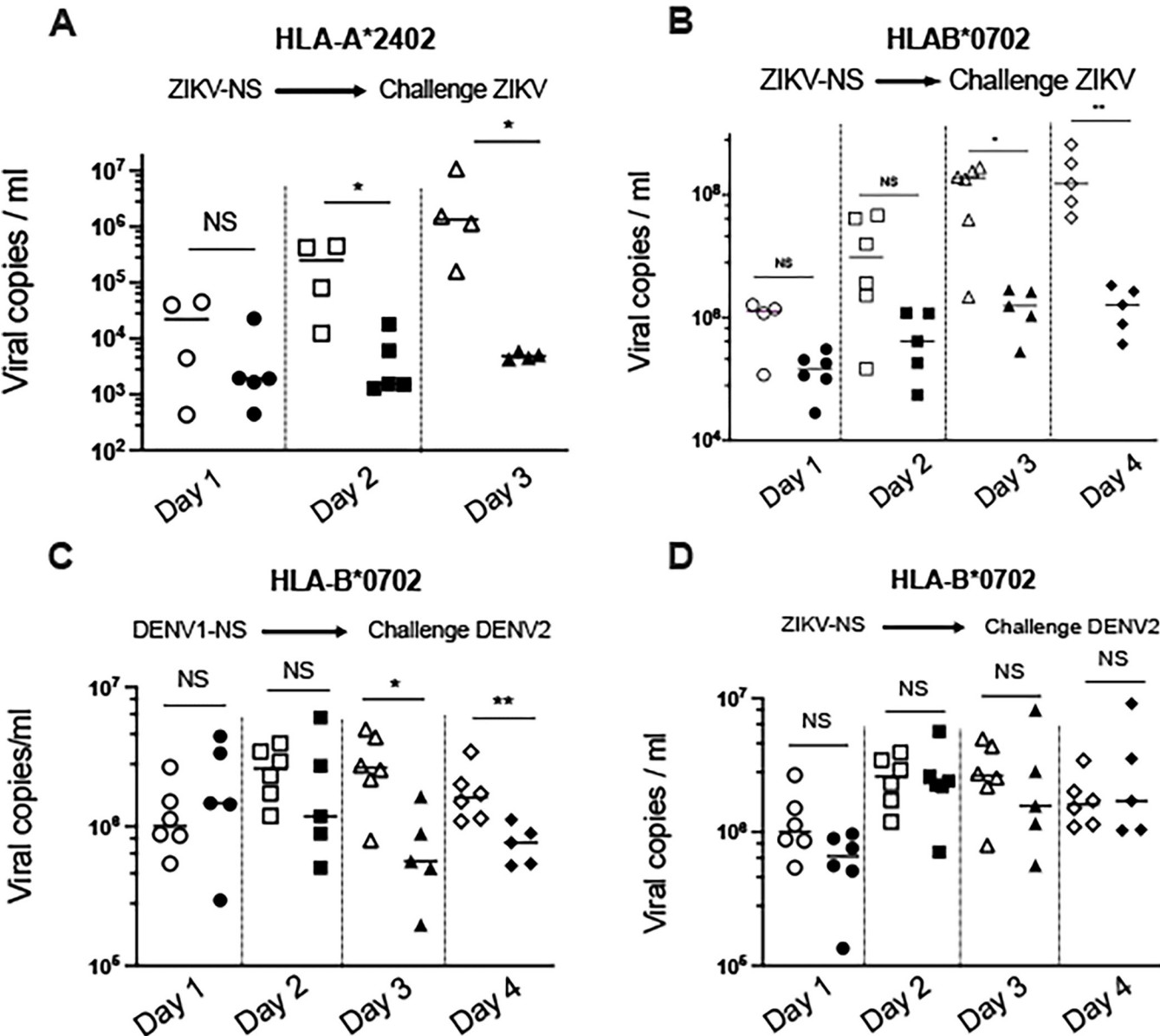

**Fig 5. Immune protection induced by immunization with the 704/DNA encoding ZIKV-NS or DENV1-NS poly-epitope.** Each panel is representative of 3 independent experiments. (A) Five and four HLA-A*2402 transgenic mice were immunized with the plasmid DNA encoding ZIKV-NS Poly-epitope (filled symbols) or the control plasmid (open symbols), respectively, and challenged with ZIKV. (B) Five and six HLA-B*0702 transgenic mice were immunized with 704/DNA encoding ZIKV-NS poly-epitope (filled symbols) or the control plasmid (open symbols), respectively, and challenged with ZIKV. (C) Five and six HLA-B*0702 transgenic mice were immunized with 704/DNA encoding DENV1-NS poly-epitope (filled symbols) or the control plasmid (open symbols), respectively, and challenged with DENV2. (D) Six HLA-B*0702 transgenic mice were immunized with the 704/DNA encoding ZIKV-NS poly-epitope (filled symbols) or the control plasmid (open symbols), respectively, and challenged with DENV2. For the immunizations, the HLA-A*2402 transgenic mice received 2 intradermic injections of 50mg ZIKV-NS or control plasmid, followed by *in vivo* electroporation, whereas the HLA-B*0702 transgenic mice received 2 intramuscular injections of 50mg 704/based DNA encoding ZIKV-NS, DENV1-NS or DNA control. Two immunizations were performed at three-week interval, at day 0 and day 21. For challenge with the viruses, one day prior challenge, mice received one intraperitoneal inoculation of 2mg of anti-IFNAR antibody (MAR1-5A3) and were challenged at day 36 with the ZIKV by Intraperitoneal injection of $10^3$ pfu Zika FG_15G strain) (A, B), or with the DENV2 virus by retro-orbital injection of $10^6$ pfu DENV2 (A0824528 IP strain) (C, D). The y axis represents the number of viral DNA copies/ml (log scale). Lines represent means and SEM. Differences between mice immunized with the ZIKV-NS or DENV1-NS vaccines and the CT vaccines were evaluated using the non-parametric Mann-Whitney U-test (* $p<0.05$, ** $p<0.01$).

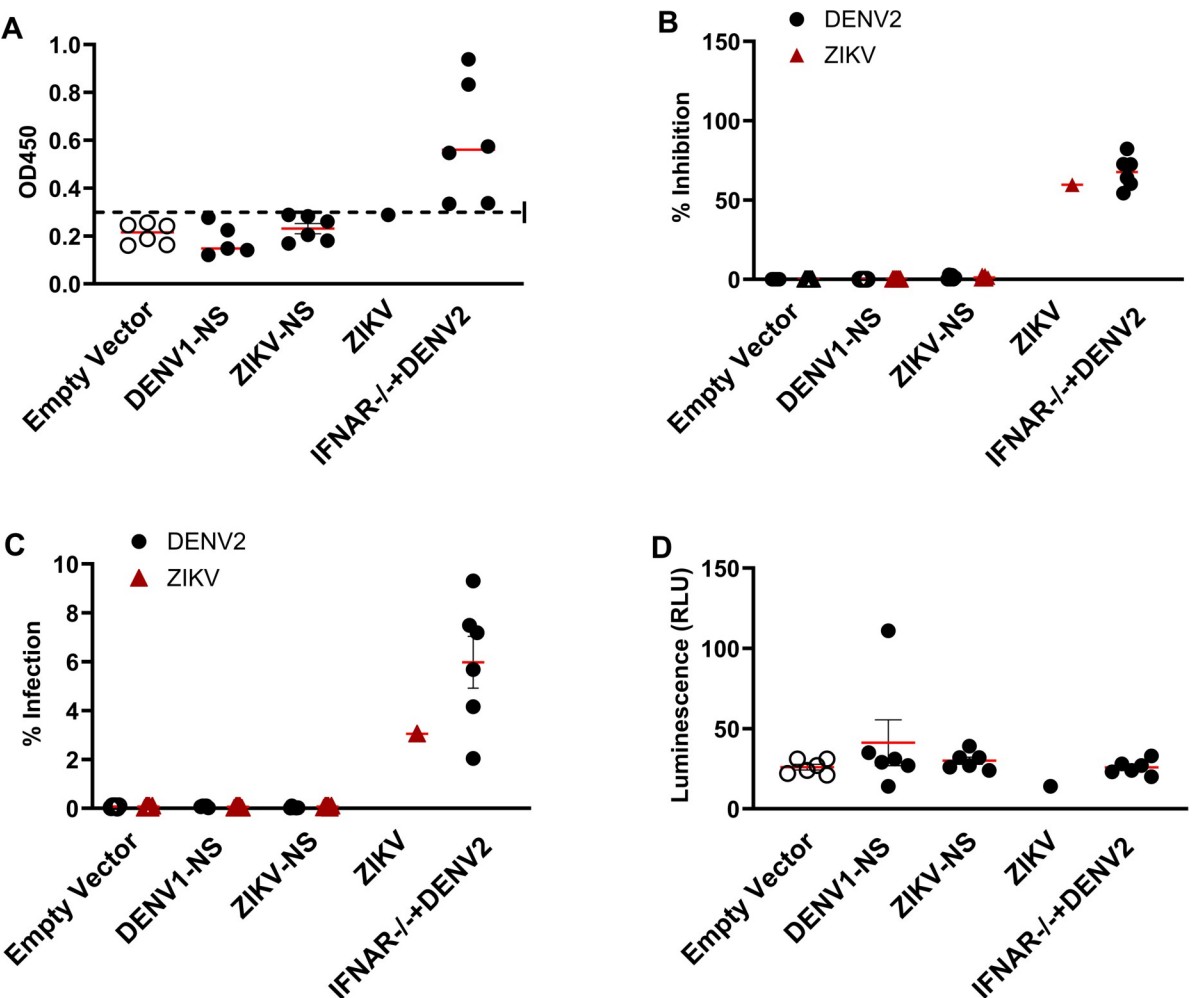

**Fig 6. Antibody response after DENV1-NS or ZIKV-NS vaccination.** Sera from five and six HLA-B*0702 transgenic mice which were immunized with 704/DNA encoding DENV1-NS or ZIKV-NS (filled symbols) or the empty vector (open symbols). IFNAR-/- mice challenged with DENV2, and ZIKV infected mice were used. (A) DENV-NS3 antibodies were detected by ELISA, (B) neutralizing antibodies were detected using a flow cytometry-based assay (DENV2, black dot; ZIKV, red triangle), (C) antibody-dependent enhancement assay (DENV2, black dot; ZIKV, red triangle), (D) Antibody-dependent cellular cytotoxicity assay with DENV2-infected cells as target cells. Each point represents one immunized mouse. Lines represent mean and SEM.

detected following DENV2 and ZIKV infection (Fig 6B). No DENV2-neutralizing antibodies were detected in the serum of animals vaccinated with 704 formulating either the DENV1-NS, ZIKV-NS poly-epitopes. Likewise, significant levels of neutralizing antibodies to ZIKV were detected following ZIKV infection but not after immunization with 704/DENV-NS or 704/ZIKV-NS poly-epitopes. Next, we ruled out the possibility of the induction of antibodies that could mediate ADE, or effector functions such as antibody-dependent cellular cytotoxicity (ADCC). As positive controls for these assays, a serial dilution of pooled serum from IFNAR$^{-/-}$ mice was used, showing a significant ADE or ADCC activity against DENV2 and ZIKV infection (S3 Fig). No ADE or ADCC activities were detected in the serum of animals vaccinated with either the DENV1-NS or ZIKV-NS poly-epitopes formulated with 704 (Fig 6C and 6D). These results confirm that vaccination with either the DENV1-NS or ZIKV-NS poly-epitope formulated with 704 does not induce protective or enhancing antibody responses.

## Discussion

Several vaccine candidates against ZIKV have been developed and are currently in phase I/II clinical trials [56,57]. However, most target prM-E and are designed to induce neutralizing Ab responses, with only a few targeting non-structural proteins to stimulate protective CD8+ T cell responses [32,58]. With the objective to prevent antibody enhancement, and the associated potential increased risk of severe disease following DENV infection after ZIKV vaccination, our approach was to induce a strong CD8+ T cell activation without inducing neutralizing or sub-neutralizing antibodies. To this end, we utilized the 704/DNA vaccine platform that has been shown to promote protective immunity against ZIKV infections, through antibodies targeting prM-E of ZIKV [46]. This platform allowed us to evaluate the ability of a poly-epitope derived from NS proteins from ZIKV to elicit protective T-cell mediated responses to ZIKV infection. Results show that immunization of HLA-A*2402 and -B*0702 transgenic mice with 704/DNA encoding ZIKV-NS induces a potent T cell response and a significant protection against ZIKV infection.

While the HLA class I transgenic mouse model is a model of choice for studying T cell response against various antigens, including viral antigens, it does not completely reflect the situation in humans. HLA class I transgenic mice usually express a single human histocompatibility allele, this restricts the number of epitopes to those presented by this HLA class I allele. For instance, among the large number of CD8-specific epitopes expressed by ZIKV and identified in humans, including numerous epitopes in the capsid and the NS1 proteins, only a few epitopes are presented by the HLA-A*2402 or -B*0702 molecules (Table 1 and Fig 2). Even with this limitation, quantification by ELISpot of the T cell response to individual peptides enabled us to identify a significant number of T cell epitopes from ZIKV-NS or DENV1-NS polyepitopes. It is important, however, to note that in these transgenic mice, the class II molecules are of murine origin, which highlights the limit of our study in that both CD4+ and CD8 + T cells have been shown to mediate protective immunity to DENV and ZIKV infection in humans [23,32,59].

In the ZIKV-NS and DENV1-NS poly-epitopes, most if not all the antigenic peptides identified in this study in the HLA-A*2402 and HLA-B*0702 transgenic mice match epitopes previously identified in humans [11,22,48,50]. Even if we did not identify the type of effector T cells responding to antigenic peptides, phenotypic analyses of the CD8 T cell compartment induced upon ZIKV-NS immunization in 2 different HLA class I backgrounds reveals an increase in the frequency of effector memory (KLRG1+ and CD62Llo) CD8 T cells. Moreover, the fact that all these antigenic peptides derived from the poly-epitopes match CD8 epitopes previously identified in humans strongly supports the fact that these peptides contain CD8+ T cell epitopes restricted by HLA class I molecules. This is the case for peptides ZV $_{2352}$ and ZV $_{2388}$ or ZV $_{1722}$ and ZV $_{2909}$ made up of HLA-A*2402- or -B*0702-restricted CD8+ T cell epitopes, respectively [50]. This is also true for peptides ZV $_{1702}$ and ZV $_{1722}$ that comprise 9- and 10-mer epitopes, respectively, previously identified in IFN-α/βR$^{-/-}$ HLA-B*0702 transgenic mice [38]. However, several 15-mer peptides, such as ZV $_{1826}$, ZV $_{2829}$ and ZV $_{1790}$, identified in the HLA-A*2402 transgenic mice, contain short peptides previously identified in IFN-α/βR$^{-/-}$ HLA-B*0702 transgenic mice, suggesting that these 15-mer peptides are composed of promiscuous or overlapping epitopes [40].

Strikingly, while the number of antigenic epitopes expressed by the ZIKV-NS differs greatly according to the HLA class I alleles, with 16 epitopes identified in HLA-A*2402 transgenic mice versus 5 epitopes identified in HLA-B*0702 transgenic mice, both types of transgenic mice are protected against ZIKV infection after vaccination with ZIKV-NS. Likewise, we have previously shown that vaccination of HLA-A*2402 and HLA-B*0702 transgenic mice with the

DENV1-NS polyepitope, which expresses five and eight epitopes restricted by these alleles, respectively, induced a significant protection against DENV1 infection [49]. Altogether, these results strongly suggests that a low number of CD8 epitopes contained in the NS poly-epitopes is sufficient to elicit a protective T-cell mediated immunity.

In humans, many ZIKV epitopes contain sequences shared between ZIKV and DENV. In contrast, in HLA-A*2402 and HLA-B*0702 mice only a small number of these CD8 T cell epitopes contain such conserved sequences between ZIKV and DENV. This raises the question of whether a limited number of CD8 epitopes with similar sequences between 2 closely related viruses can induce cross-protection. To answer this question, we assessed the ability of ZIKV-NS and DENV1-NS poly-epitopes that possess either a low or high number of epitopes cross-reactive with DENV2 to induce a cross-protection. Results show that HLA-B*0702 mice immunized with the ZIKV-NS poly-epitope, which contains five epitopes from ZIKV, of which only two have sequences shared with DENV2, are protected against ZIKV infection, but not against DENV2 infection. Conversely, immunization of HLA-B*0702 mice with the DENV1-NS poly-epitope, which contains eight epitopes from DENV1, of which six have similar sequences with DENV2 peptides, induces a significant protection against both DENV1 and DENV2 infection [49] (and this study). Altogether, these results show that at least six CD8 T cell epitopes with similar sequences shared between 2 close viruses are sufficient to elicit a cross-protective immunity.

We demonstrated that immunization of HLA-B*0702 mice with DENV1-NS results in the activation of cross-reactive T cells that recognize with the same magnitude both DENV1- and DENV2-derived peptides [49], and in the immune protection against DENV1 [49] and DENV2 (this study), Due to the limited number of HLA class I transgenic mice available, a limitation of the current study is that we could not assess the T cell activity to DENV peptides in mice immunized with ZIKV-NS nor to ZIKV peptides in mice immunized with DENV1-NS, nor to peptides with the invariant sequence between DENV and ZIKV.

Since immunization with ZIKV-NS or DENV1-NS does not induce the production of neutralizing or sub-neutralizing antibodies against ZIKV or DENV, nor antibodies mediating cellular cytotoxicity, our results clearly show that the immune protection against ZIKV or DENV infection directly depends on the activation of T cells. This beneficial role of T cells is in line with other studies showing that CD8 T cells induced by a NS3-based T cell vaccine prevents ZIKV infection and fetal damage in mice [36,58].

Immunization with a T cell directed vaccine for flaviviruses has several advantages. First, it could elicit a long-lasting immunity against infected cells by activating memory CD8 and CD4 T cells [60]. Further studies in mice are needed to determine the optimal conditions of immunization to stimulate these memory CD4 and CD8 T cells. In addition, the advantage of a vaccine containing NS-derived T cell epitopes resides in its capacity to strongly activate CD8 T cells that could prevent ADE, as shown in several animal models [61,62]. In endemic regions where there is high simultaneous circulation of multiple DENV serotypes and/or ZIKV, these NS-derived T cell vaccines could induce cross-protective immunity in the absence of potential enhancing antibodies. Altogether, these results open up avenues for the development of multivalent DNA vaccines eliciting T cell responses against multiple flaviviruses conferring protection from infection and disease.

## Supporting information

**S1 Table. ZIKV-NS and DENV1-NS T cell epitopes.** Amino acid sequences in red represent amino acids that differ from the antigenic sequence of the immunizing poly-epitope. Underlined sequences represent amino acid sequences having at least 80% sequence identity with the

immunizing ZIKV-NS or DENV1-NS poly-epitopes. Numbers in brackets indicate the percentages of identity with the antigenic peptides from the immunizing poly-epitopes.
(DOCX)

**S1 Fig. Gating strategy for phenotypic analyses of CD3+ CD8+ T cells in HLA-A\*2402 and HLA-A\*0201 transgenic mice immunized with the 704 complexed ZIKV-NS poly-epitope (ZIKV-NS) or the empty vector (CT).** HLA-A\*2402 mice immunized with ZIKV-NS or control vector, were tested at day 10 after the secondary immunization (5 mice per group), and at day 24 after the secondary immunization (3 mice per group). HLA-A\*0201 mice were immunized with ZIKV-NS or control vector and tested at day 10 and day 24 after the secondary immunization (3 mice per group).
(PDF)

**S2 Fig. Quantification of the T cell response in HLA-A\*0201 and HLA-B\*3501 transgenic mice by IFN-γ and Granzyme B ELISpot assay.** Closed and open symbols represent the animals immunized with 704 formulated DENV1-NS and the control plasmid, respectively. All the animals were immunized by intramuscular injection (2x50μg) at day 0 and day 21, and spleen cells were tested for IFN-γ or Granzyme B secretion by ELISpot assay 10 days after the second injection. Individual mice were tested in parallel with different peptides at 2μg/ml and with concanavalin A at 5μg/ml, final concentration. Peptides p30, p53, p56, and p49, p50, p51 have been characterized previously (ref 49). Lines represent mean and SEM. (n varied between 4 and 6 mice/group except for HLA-B\*3501 transgenic mice with n = 3 and 1 for DENV1-NS and control plasmid, respectively. Differences between mice immunized with DENV1-NS and the control plasmid were evaluated using non-parametric Mann-Whitney U-test (\*p<0.05, \*\*p<0.01).
(TIF)

**S3 Fig. Optimization of assays to evaluate the humoral immune response against DENV.**
(A) flow cytometry-based neutralization assay measuring the percentage of infection. Vero CCL were in vitro infected with DENV-2 (red dot line) and ZIKV (black line) in presence or absence of serially diluted IFNAR-/- DENV2 challenged and ZIKV infected sera (B) Antibody-dependent enhancement assay. U937 cells were in vitro infected with DENV-2 (red dot line) and ZIKV (black line) in presence or absence of serially diluted IFNAR-/- DENV2 challenged and ZIKV infected sera. (C) Antibody-dependent cellular cytotoxicity assay. Jurkat reporter cells (effector cells) were incubated with Raji cell (target cells) in presence or absence of rituximab (anti-CD20) mAb. The data is shown as luminescence (RLU).
(PDF)

## Acknowledgments

We are very grateful to Alessandro Sette for providing the HLA-B\*07:02/IFNAR1 transgenic mice colony. We also thank Dorian Caudal and Camille Chauvin for their contributions to the vaccination procedures.

## Author Contributions

**Conceptualization:** Claude Roth, Tineke Cantaert.

**Data curation:** Claude Roth, Tineke Cantaert.

**Formal analysis:** Claude Roth, Laurine Levillayer, Sokchea Lay, Hoa Thi My Vo, Tineke Cantaert.

**Funding acquisition:** Anavaj Sakuntabhai.

**Investigation:** Claude Roth, Laurine Levillayer, Sokchea Lay, Hoa Thi My Vo.

**Methodology:** Claude Roth, Bruno Pitard, Laurine Levillayer, Sokchea Lay, Hoa Thi My Vo, Tineke Cantaert.

**Project administration:** Bruno Pitard, Tineke Cantaert, Anavaj Sakuntabhai.

**Supervision:** Claude Roth, Anavaj Sakuntabhai.

**Validation:** Claude Roth, Laurine Levillayer, Hoa Thi My Vo, Tineke Cantaert.

**Visualization:** Bruno Pitard, Sokchea Lay.

**Writing – original draft:** Claude Roth.

**Writing – review & editing:** Claude Roth, Bruno Pitard, Tineke Cantaert.

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
