## [Decision Letter · Decision Letter 0]

13 May 2024

Dear Dr Roth,

Thank you very much for submitting your manuscript "Zika virus T-cell based 704/DNA vaccine promotes protection from Zika virus infection in the absence of neutralizing antibodies" for consideration at PLOS Neglected Tropical Diseases. As with all papers reviewed by the journal, your manuscript was reviewed by members of the editorial board and by several independent reviewers. In light of the reviews (below this email), we would like to invite the resubmission of a significantly-revised version that takes into account the reviewers' comments. 

The reviewers agree this is a very important topic however they have a number of major concerns meaning that we are unable to consider publication of the manuscript in its current form. In particular, both reviewers have highlighted that not enough work has been carried out to allow understanding of the phenotype and function of the T cell response, given that the premise of the work is to highlight T cell inducing vaccines for Zika. In addition, further experimental work addressing the cross reactivity of the response is recommended, and consideration of using IFNg KO mice to explore the efficacy. If the authors are able to complete such further work we are willing to reconsider for publication but please note this is not a guarantee.

We cannot make any decision about publication until we have seen the revised manuscript and your response to the reviewers' comments. Your revised manuscript is also likely to be sent to reviewers for further evaluation.

Sincerely,

Susanna Jane Dunachie

Guest Editor

Andrea Marzi

Section Editor

Thank you for your recent submission. The reviewers agree this is a very important topic however they have a number of major concerns meaning that we are unable to consider publication of the manuscript in its current form. In particular, both reviewers have highlighted that not enough work has been carried out to allow understanding of the phenotype and function of the T cell response, given that the premise of the work is to highlight T cell inducing vaccines for Zika. In addition, further experimental work addressing the cross reactivity of the response is recommended, and consideration of using IFNg KO mice to explore the efficacy. If the authors are able to complete such further work we are willing to reconsider for publication but please note this is not a guarantee.

Reviewer's Responses to Questions

**Key Review Criteria Required for Acceptance?**

**Methods**

-Are the objectives of the study clearly articulated with a clear testable hypothesis stated?

-Is the study design appropriate to address the stated objectives?

-Is the population clearly described and appropriate for the hypothesis being tested?

-Is the sample size sufficient to ensure adequate power to address the hypothesis being tested?

-Were correct statistical analysis used to support conclusions?

-Are there concerns about ethical or regulatory requirements being met?

Reviewer #1: 1. In the Introduction the authors allude to the concept of original antigenic sin but they do not necessarily describe the term. In the context of DENV this is important and should be mentioned in the intro.

2. The authors state that “ZIKV-CD8+ T cells are an important correlate of protection ...” however up to now this statement has not been clinically proved. The authors should consider re-phrasing the sentence given that the best correlates of protection continue to be the presence of neutralising antibodies. I would also argue that the role of CD4+ T cells is a bit clearer than the authors state. For example, studies by Wen et al. (CD4+ T Cells Cross-Reactive with Dengue and Zika Viruses Protect against Zika Virus Infection) and others clearly demonstrate the role of CD4+ T cells in the absence of neutralizing antibodies. Moreover, recent work from Carlos Sariol’s lab show that depleting CD4+ T cells is associated with increases in DENV and ZIKV viremia and describe a role for CD4 helper T cells.

3. For the immunizations I am surprised that there is no empty vector control (i.e., a 704-complexed pVAX1 plasmid without any antigen). Do the authors have access to this construct? I suspect there would be very little responses but that information would be important to have and to show in the manuscript.

4. Given the authors want to demonstrate cross-protection did the researchers think about immunizing with the poly-epitope Zika vaccine and then testing splenocytes against dengue peptides or vice versa? Figures 1 and 2 show ZIKV vaccination and/or DENV vaccination with splenocytes being tested on antigen-specific peptides. Have the authors thought about testing the splenocytes against invariant peptides across the dengue serogroup and others? If material is available this should be a consideration especially since the response is directed to a specific peptide. This also would provide concrete proof for the utility of the vaccine rather than stating ZIKV vaccination resulted in responses to these peptides which have 80% sequence identity to DENV2.

5. Given the goal of the work is to induce potent CD8+ T cell responses through vaccination I am surprised that there was no immunophenotyping performed. If CD8 T cells were induced through vaccination were they degranulating? Were the T cell responses polyfunctional? This is a significant hole in the work and if material is still available from these animals I would strongly suggest looking at multiple T cell markers including CD4, CD8, IFNg, IL2, CD107a to name a few. This data would also provide the authors with a potential mechanism to explain the decreased viremia they observe post vaccination.

Reviewer #2: The objectives are clearly defined but the results are not completed.

The sample size could be increased to make the data stronger

**Results**

-Does the analysis presented match the analysis plan?

-Are the results clearly and completely presented?

-Are the figures (Tables, Images) of sufficient quality for clarity?

Reviewer #1: 1. In Table 1, the columns state “Total no. of SFCs” and “Avg no. of SFCs (positive donors)” what is the difference? I though all SFCs were from donors infected with ZIKV. I may have misunderstood but that is not very clear. What was the SFCs for the negative control (i.e., DMSO)? Was that value considered in the analysis? It would be helpful to the reader if a few sentences were added to the methods in the ELISpot section which state how the SFCs were derived. In review of Table 1, it does seem that some of the epitopes in the vaccine only induce a maximum of 50% T cell responses. For example, with NS1.2 989-1003 (HSDLGYWIESEKNDT) the total spots were 615 and only 307 were from positive donors (49.9%).

2. For some of the panels in Figure 1, medium and ConA are shown and for others it is not. The authors should keep the different panels consistent. In addition, the figure legend should state that the x-axis represents the epitopes for which a response was made. Also, why was data for capsid and NS1 not shown? Given the nature of the vaccine, the authors should consider showing all the ELISpot data from all the peptides tested rather than a few peptides as seen in Figure 1.

Reviewer #2: Major clarifications should be made in the manuscript

**Conclusions**

-Are the conclusions supported by the data presented?

-Are the limitations of analysis clearly described?

-Do the authors discuss how these data can be helpful to advance our understanding of the topic under study?

-Is public health relevance addressed?

Reviewer #1: The manuscript would have greatly benefited from two key experiments: (1) demonstrating cross-reactivity either by ELISpot using invariant peptides or through killing assays with T cell lines directed against a specific peptide and an APC line to an invariant peptide, AND (2) cellular phenotyping. The authors conclude about the protective role of T cells but do not surmise a potential mechanism which through immunophenotyping they would have obtained functional data. The manuscript is important however more work needs to be done to be published.

Reviewer #2: The limitations of the study were not clearly defined in the manuscript

**Editorial and Data Presentation Modifications?**

Reviewer #1: 1. Line 25 Page 2 and Line 7 Page 3, Aedes should be italicised.

2. Line 23 Page 3 envelope is spelled incorrectly, missing an “e”.

3. Page 4, the authors should state that ADE stands for antibody dependent enhancement.

4. In Figure 4, state that the y-axis is log-transformed on the figure or in the legend.

Reviewer #2: (No Response)

**Summary and General Comments**

Reviewer #1: Figure 2 clearly shows that certain regions of the flavivirus genome are “hotspots” for T cell activity which begs the question why add capsid and NS1.

Reviewer #2: The manuscript PNTD-D-24-00451 titled “Zika Virus T cell based 704/DNA Vaccine promotes protection from Zika virus infection in the absence of neutralizing antibodies” submitted by Roth C, aims to develop a T cell-based vaccine against ZIKV infection. The authors utilized a DNA vaccine platform using the tetrafunctional amphiphilic block copolymer 704. They evaluated the immunogenicity and efficacy of the DNA vaccine encoding for ZIKV Non-Structural (NS) polyepitope using two different HLA class I transgenic mice. The authors showed that the vaccination elicits a T cell response against many immunodominant ZIKV epitopes and conferred protection (based on viremia) against ZIKV but not against DENV2. However, DENV1 NS poly-epitope used as a control provided cross-protection against DENV2. This study addresses an important topic concerning the development, efficacy, and safety of T cell-based vaccines against flaviviruses as we know that unbalanced humoral immunity induced by vaccine could trigger the development of severe disease after subsequent infections. However, there are many missing points that should be addressed to emphasize and clarify the findings. 

Major comments:

Immunogenicity

As the focus of the manuscript is the characterization of a T cell-based vaccine, it is expected that the authors will explore more the T cell response induced by the vaccine. The screening of peptides using IFN�-Elispot is good but not enough for the immunogenicity of the vaccines. For a T cell-based vaccine, it is important to know what is the phenotype/ function of the T cells elicited by the vaccine: Activation markers, Cytokine production, cytolytic activity? The immunogenicity here is incomplete.

The authors compared the immunogenicity in transgenic (Tg) WT mice vs. Tg IFN-��R-/- mice. It would also be informative to compare the T cell responses (cytokine profile, activation etc..) induced by the vaccine in both mouse strains (WT vs. IFNAR-/-). Are the profiles of CD8 T cells different after vaccination in both strains?

Efficacy

Similarly, to the immunogenicity, it is not clear why the authors are not using the Tg IFN-��R-/- mice for efficacy instead of Tg WT mice.

It is unclear why the authors chose to quantify viral DNA copies instead of viral RNA copies. In the methodology section for the quantification of viral loads (page 7), details regarding the targeted genes are missing to ensure that DNA copies from the vaccination will not interfere with the quantification of viral DNA. How long does the DNA vaccine last in tissues or in periphery?

For an efficacy study using mice, the viremia is not enough to claim protection. What is the viral load in other organs, such as brain, testes, liver spleen? For ZIKV and DENV1,2. 

How many experiments are represented for each panel/figure? The number of mice could be increased for some experiments.

Minor comments:

Figure 1: Missing controls (ConA and Medium) for some panels

Figure 4: Day 4 is missing for panel A.

Figure 5: What is exactly ZIKV mentioned before IFNAR-/- +DENV2? If each point represents one immunized mouse, why ZIKV has only one point? Is it just one mouse? Why? Please clarify.

Table 1 summarizes the immunodominant epitopes, however the authors should clarify how these epitopes were obtained. What strain of ZIKV was used? What approach? Prediction? Overlapping? 

Page 15, line 1-7: what figure is being analyzed?

The authors only mentioned electroporation in Figure 4 legend but not in the methodology section page 6, Is the electroporation only used for efficacy? If yes, why?

Why the viruses (ZIKV, DENV) are inoculated using different routes? Intraperitoneal vs. retroorbital.

The neutralization activity of the positive control group seems low, this could be due to the representation in % of inhibition instead of NT50. The NT50 is more appropriate to evaluate the neutralization activity.

PLOS authors have the option to publish the peer review history of their article (what does this mean?). If published, this will include your full peer review and any attached files.

Reviewer #1: No

Reviewer #2: No
---

## [Decision Letter · Decision Letter 1]

9 Sep 2024

Dear Dr Roth,

Thank you very much for submitting your manuscript "Zika virus T-cell based 704/DNA vaccine promotes protection from Zika virus infection in the absence of neutralizing antibodies" for consideration at PLOS Neglected Tropical Diseases. As with all papers reviewed by the journal, your manuscript was reviewed by members of the editorial board and by several independent reviewers. The reviewers appreciated the attention to an important topic. Based on the reviews, we are likely to accept this manuscript for publication, providing that you modify the manuscript according to the review recommendations. 

Please could you address the minor issues raised by Reviewer 1, plus also carefully review the manuscript for any remaining typographical errors, including the following examples:

- Results line 12 page 10: There are only 19 peptides in Table 1 instead of 20 peptides as mentioned in the text. Please correct.

- Figure 5 and 6 are misplaced.

- The title page 14, line 8-9 mentioned only ZIKV-NS polyepitope, although DENV1-NS is also described in the text result.

Sincerely,

Susanna Jane Dunachie

Guest Editor

Andrea Marzi

Section Editor

Please could you address the minor issues raised by Reviewer 1 plus check the following, below.

Carefully review the manuscript for any remaining typographical errors.

For example, results line 12 page 10: There are only 19 peptides in Table 1 instead of 20 peptides as mentioned in the text. Please correct.

Figure 5 and 6 are misplaced.

The title page 14, line 8-9 mentioned only ZIKV-NS polyepitope, although DENV1-NS is also described in the text result.

If these are addressed adequately we should be in a position to accept the manuscript.

Reviewer's Responses to Questions

**Key Review Criteria Required for Acceptance?**

**Methods**

-Are the objectives of the study clearly articulated with a clear testable hypothesis stated?

-Is the study design appropriate to address the stated objectives?

-Is the population clearly described and appropriate for the hypothesis being tested?

-Is the sample size sufficient to ensure adequate power to address the hypothesis being tested?

-Were correct statistical analysis used to support conclusions?

-Are there concerns about ethical or regulatory requirements being met?

Reviewer #1: In the revised manuscript the authors have addressed the issues raised.

Reviewer #2: (No Response)

**Results**

-Does the analysis presented match the analysis plan?

-Are the results clearly and completely presented?

-Are the figures (Tables, Images) of sufficient quality for clarity?

Reviewer #1: In the revised manuscript the authors have addressed the issues raised.

Reviewer #2: (No Response)

**Conclusions**

-Are the conclusions supported by the data presented?

-Are the limitations of analysis clearly described?

-Do the authors discuss how these data can be helpful to advance our understanding of the topic under study?

-Is public health relevance addressed?

Reviewer #1: In the revised manuscript the authors have addressed the issues raised.

Reviewer #2: (No Response)

**Editorial and Data Presentation Modifications?**

Reviewer #1: (No Response)

Reviewer #2: (No Response)

**Summary and General Comments**

Reviewer #1: Only minor changes to enhance the clarity of the revised manuscript:

Page 3

Line 7. Change the sentence “ZIKV is transmitted …” to “ZIKV transmission occurs via the bite of an infected Aedes mosquito and in some cases through human sexual contact or the maternal fetal route (2).

Line 19 – Remove “More specifically”

Line 21 – change the sentence to “…and DENV4, sequence identity is 67-68% for the NS3 and NS5 proteins (10)”.

Lines 35-37 – I agree with the authors that the levels of pre-existing antibodies influence secondary infections, but I would say that it’s also the quality of those antibodies as highlighted in reference 19 that is important. A minor change rewriting of this sentence is needed.

Page 4

Line 21 – change “protecting” to “protective”

Line 28 – change “CD4 T cell” to CD4 T cells

Page 5

Line 14 – add “the ZIKV proteome”

Page 14

Line 23 – remove “in”

Reviewer #2: (No Response)

PLOS authors have the option to publish the peer review history of their article (what does this mean?). If published, this will include your full peer review and any attached files.

Reviewer #1: Yes: Krishanthi Subramaniam

Reviewer #2: No

Figure Files:

Data Requirements:

Reproducibility:

References

---

## [Editor Report · Decision Letter 2]

3 Oct 2024

Dear Dr Roth,

We are pleased to inform you that your manuscript 'Zika virus T-cell based 704/DNA vaccine promotes protection from Zika virus infection in the absence of neutralizing antibodies' has been provisionally accepted for publication in PLOS Neglected Tropical Diseases.

Best regards,

Susanna Jane Dunachie

Guest Editor

Andrea Marzi

Section Editor

---

## [Editor Report · Acceptance letter]

9 Oct 2024

Dear Dr Roth,

We are delighted to inform you that your manuscript, "Zika virus T-cell based 704/DNA vaccine promotes protection from Zika virus infection in the absence of neutralizing antibodies," has been formally accepted for publication in PLOS Neglected Tropical Diseases.

Best regards,

Shaden Kamhawi

co-Editor-in-Chief

Paul Brindley

co-Editor-in-Chief
